# Robotic-Assisted Bronchoscopy: A Comprehensive Review of System Functions and Analysis of Outcome Data

**DOI:** 10.3390/diagnostics14040399

**Published:** 2024-02-12

**Authors:** Renan Martins Gomes Prado, Joseph Cicenia, Francisco Aécio Almeida

**Affiliations:** 1School of Medicine, Center of Health Sciences, State University of Ceara, Fortaleza 60714-903, Brazil; 2Department of Pulmonary Medicine, Cleveland Clinic, Cleveland, OH 44195, USA

**Keywords:** robotic bronchoscopy, robotic-assisted bronchoscopy, peripheral bronchoscopy, guided bronchoscopy

## Abstract

The past two decades have witnessed a revolutionary era for peripheral bronchoscopy. Though the initial description of radial endobronchial ultrasound can be traced back to 1992, it was not until the mid-2000s that its utilization became commonplace, primarily due to the introduction of electromagnetic navigation (EMN) bronchoscopy. While the diagnostic yield of EMN-assisted sampling has shown substantial improvement over historical fluoroscopy-assisted bronchoscopic biopsy, its diagnostic yield plateaued at around 70%. Factors contributing to this relatively low diagnostic yield include discrepancies in computed tomography to body divergence, which led to unsuccessful lesion localization and resultant unsuccessful sampling of the lesion. Furthermore, much of peripheral bronchoscopy utilized a plastic extended working channel whose tips were difficult to finely aim at potential targets. However, the recent introduction of robotic-assisted bronchoscopy, and its associated stability within the peripheral lung, has ignited optimism for its potential to significantly enhance the diagnostic performance for peripheral lesions. Moreover, some envision this technology eventually playing a pivotal role in the therapeutic delivery to lung tumors. This review aims to describe the currently available robotic-assisted bronchoscopy technologies and to discuss the existing scientific evidence supporting these.

## 1. Introduction

Bronchoscopy for the diagnosis of nodules and other lung lesions has been used at least since the early 1970s. However, the sensitivity of conventional bronchoscopy, generally using fluoroscopy, varies from 14% to 63%, depending on the size and location of the lesion and the presence or absence of the bronchus sign [1,2]. Although percutaneous transthoracic needle aspiration (TTNA) has an excellent diagnostic sensitivity of around 90% [3], complications are reported in up to 43% of cases [3,4]. Pneumothorax rates of 12–45% have been reported of which 2–15% undergo thoracostomy tube placement [5]. In addition, staging of the mediastinum and hilar regions with endobronchial ultrasound (EBUS) cannot be performed simultaneously [6].

The last two decades have been revolutionary for peripheral bronchoscopy. Even though the radial probe (RP)-EBUS was initially described in 1992 [7], its routine use only began in the mid-2000s with the introduction of electromagnetic navigation (EMN) bronchoscopy [8]. Despite the excitement with the growth of peripheral bronchoscopy, a meta-analysis in 2012 of the technologies in use up to that point (EMN, virtual bronchoscopy, RP-EBUS, ultrathin bronchoscope, and guide sheath) demonstrated a diagnostic yield of around 70% regardless of the technology used [9]. Since then, 70% has been considered by the scientific community as the diagnostic yield to be surpassed with the evolution of existing techniques or the introduction of new technologies. Computed tomography (CT)-to-body divergence and diagnostic drop-off (discrepancy between successful lesion localization and obtaining a sample) are challenges that have been considered significant factors in existing technologies’ relatively low diagnostic yield [10].

The recent introduction of robotic-assisted bronchoscopy (RAB) has brought significant optimism in resolving at least the diagnostic drop-off. Moreover, although a recent update of the meta-analysis cited above has not shown an improvement in the diagnostic yield of existing technologies as a whole [11], RAB was described as having an average diagnostic yield of almost 77% in its initial studies. The objectives of this review will be to describe how RAB technologies currently work and to discuss the current scientific evidence.

## 2. Robotic-Assisted Bronchoscopy Technologies

Currently, there are three platforms available for RAB. The Ion™ platform (Intuitive Surgical, Sunnyvale, CA, USA), which has a history in robotic-assisted surgery (Figure 1A), the Monarch™ platform (Ethicon, Sunnyvale, CA, USA) (Figure 1B), and the Galaxy System™ (Noah Medical, Sunnyvale, CA, USA) (Figure 1C), which was approved for use by the Food and Drug Administration (FDA) in March 2023.

### 2.1. Navigation 

Despite the general robotic similarity, the Ion™, Monarch™, and Galaxy™ platforms have specific features that make them significantly different (Table 1). Unlike traditional EMN-based systems, the Ion™ platform uses shape-sensing technology, used for the first time in bronchoscopy. This optical fiber shape-detection technology can measure the catheter’s form hundreds of times per second. In this way, the platform may precisely determine the bronchoscope’s shape and location. There is no known interference with fluoroscopy or any metal according to the company’s information and our experience. The Monarch™ and Galaxy™ systems, on the other hand, use electromagnetic navigation technology present in the market since the mid-2000s. Therefore, these technologies can be prone to metal interference. Ion™ recently incorporated a system to address CT-to-body divergence (divergence in the expected and actual location of the target lesion due to discrepancies in lung anatomy between the preprocedural CT and the actual bronchoscopic procedure) by incorporating a target-location-updating function utilizing digital tomography. At the moment, this function can only be enabled for use with the Cios Spin^®^ Mobile C-arm (Siemens, Erlangen, Germany). The Galaxy™ platform also has similar target-location-updating technology already incorporated into its system and available to use with most fluoroscopy systems currently in use in the United States. The availability of such technology appears to be crucial as CT-to-body divergence has been reported to occur in at least 50% of cases [12]. The Galaxy system also comes with built-in augmented fluoroscopy. As of the preparation of this article, the Monarch™ system does not yet have either of these technologies available.

### 2.2. Catheter/Bronchoscope 

The Ion™ uses a catheter with a removable camera (Figure 2A). Thus, while using the tools to collect tissue, there is no visualization of the airway (Figure 2B). The Monarch™ and Galaxy, on the other hand, can provide this visualization throughout the entire procedure (Figure 2C–E). In practice, this visualization sometimes can be very helpful to adjust the scope direction (Figure 2E) during the biopsy (Figure 2F). This platform has a sheath that the operator uncouples from the bronchoscope when in a more distal position. The articulation of both platforms is excellent (up to 180 degrees in all directions). Notably, the articulation of both generally does not change or lose flexion when any tool is inserted, unlike other technologies where the catheter or the common bronchoscope is more malleable. In theory, the “drop-off” should not occur. Based on our experience, however, this does occur on occasion, or the operator must straighten the scope, sometimes requiring slight retraction, in order to advance the sampling tool before returning the scope/catheter to its sampling direction and/or position. This ability to remain stable is considered one of the greatest virtues of robotic platforms for potential use in treating lung tumors. The Monarch™ and Galaxy™ offer irrigation and aspiration, and the Ion™ does not. Because of the presence of the working channel in addition to the built-in camera, both the Monarch™ and Galaxy™ have a slightly larger external diameter and theoretically may not be able to navigate as distally as the Ion™. Their working channel diameters are similar. A common problem with robotic scopes is that there is no tactile feedback system. Sometimes the view of the airway is lost, and it is impossible to perceive if the bronchoscope/catheter may be lacerating or even perforating it.

### 2.3. Controls 

The Ion™ control has two “buttons” on a base coupled to a small screen (Figure 3A). The latter works like a touchscreen smartphone where the operator can use their fingers to utilize its functions. The Monarch™ and Galaxy controllers resemble an Xbox video game controller (Figure 3B,C). These controllers also have irrigation and suction buttons around the index fingers. The Ion™ does not have these.

### 2.4. Display Screens

The navigation screens of the platforms are similar. All screen display changes can be made on the manual controls. On the Ion™ (Figure 2B), these changes are generally made by clicking on the small screen that is part of the manual control (Figure 3A). On the Monarch™ (Figure 2E), the changes are generally made by pressing the various buttons on the manual control while observing the screen to choose the appropriate options. The larger screen can also be used for option changes on both platforms. The Galaxy™ uses a touch screen monitor. The Ion™ estimates the distance from the lesion to the pleura, which, in theory, can help the operator avoid puncturing/biopsying the latter. The Ion™ also has a function called “cloud biopsy” in case the platform operator changes puncture positions in search of a better position. The platform then memorizes the various positions. Thus, if the pathologist is available in the procedure room, the operator can return to the position where the best-quality material was collected based on the pathologist’s information. The Galaxy, since it uses digital tomography, also has a targeting screen (Figure 4D) and incorporates tool-in-lesion (TILT+ ^TM^) into the user interface (Figure 4E).

## 3. Procedure Setting and Sedation/Anesthesia

### 3.1. Location Setting

The RAB procedure is carried out in an outpatient setting. Following the procedure’s completion, patients can return home on the same day. In our organization, patients are often discharged within an hour post-RAB. This is made possible by the minimally invasive nature of the procedure and the short recovery time involved.

### 3.2. Sedation/Anesthesia

Studies to date summarized in the literature section below report utilizing endotracheal intubation and general anesthesia for all patients undergoing RAB procedures. Neuromuscular blockade was commonly reported. Recently, there has been a documented case report involving the use of RAB under moderate sedation for treating a sizable peripheral mass [13]. We believe it to be difficult for this sedation approach to become widespread. First, the speed at which the scope moves in the airway is much slower than one can move a standard bronchoscope in the event a rapid repositioning is necessary. More importantly, most of the procedures are performed to sample small and peripheral lesions. As such, CT-to-body divergence and the development of atelectasis during the procedure are major challenges that generally require specific ventilatory management that cannot be achieved without positive pressure ventilation [14,15,16,17,18]. The first study specifically evaluating the incidence of atelectasis during bronchoscopy demonstrated that it develops as early as 3 minutes into the procedure [16]. Data are lacking on the incidence of atelectasis under conscious sedation, but we believe this probably occurs as well. Because RAB procedures are generally longer when compared to typical conscious sedation procedures, these patients are likely to receive more sedatives and/or narcotics and end up developing more atelectasis.

## 4. Literature

Several pertinent aspects warrant thoughtful consideration before delving into the pivotal studies in RAB. The size of the lesion matters since lesions larger than 20 mm have a diagnostic yield 20% higher than smaller lesions [11]. However, other aspects may also be important in the probability of providing a diagnosis. The presence of bronchus sign seems to increase the probability of diagnosis [19]. The selection bias of cases and the prevalence of malignant disease also seem impactful [11]. Finally, diagnostic yield is not uniformly defined across publications. For assessing the diagnostic accuracy of each technology, we categorized it as “strict” when the authors specified a distinct diagnosis for each procedure. “Not strict” referred to instances where the authors relied on stability in follow-up imaging to aid in determining diagnostic yield for non-diagnostic samples.

### 4.1. Ion™

Table 2 summarizes the main studies. The initial human study described an average lesion size of 14.8 mm, considered a small diameter for bronchoscopic studies [20]. The diagnostic yield described by the authors was almost 10% above the general average of all technologies mentioned initially. Another study to evaluate the platform’s safety in 67 lesions reported that nearly 90% of the lesions could be visualized with RP-EBUS, despite the bronchus sign being present in less than 40% [21]. Some tissue could be obtained in 97% of the lesions but the authors did not report their diagnostic yield. Cone-beam CT (CBCT) in combination with Ion™ was then initially evaluated in an observational study [22]. Due to the use of cone-beam CT, RP-EBUS was not used. The authors described that, with some adjustments of the catheter based on CT (CT-to-body divergence), a tool-in-lesion (TIL) was achieved in 100% of cases. The average duration of the procedure with this combination was over an hour. The prospective single-center study published by Kalchiem-Dekel and collaborators was the first study with a relatively high number of lesions [23]. An excellent follow-up for non-diagnostic cases was also described. Lesions were visualized with RP-EBUS in over 90%, even with 56% of them being less than 20 mm. Despite the diagnostic yield exceeding 80%, it is worth noting that this was 66.6% for lesions of 10 mm or smaller and 70.4% for lesions from 10.1 to 20 mm. The work by Reisenauer et al. seems to be a continuation of the safety study mentioned earlier, but the authors did not make this clear [24]. As in the original study, the diagnostic yield was not mentioned, but the authors described the diagnosis as adequate in 95% of cases, and 65% of the lesions were malignant. The diagnostic accuracy will be published after a follow-up that the authors consider appropriate. A study conducted at a single center reported on their experience with the first 200 cases (two patients had two procedures) in which CBCT and/or RP-EBUS were used [25]. A follow-up of 3 to 14 months was described, and lesions without adequate follow-up were considered “false negatives” (the authors do not specify, but it is understood to be for malignant disease). Cone-beam CT was used in all but three cases. The accuracy for malignant disease was 91.4%, with a sensitivity of 87.3%. Diagnostic yield was not used in the analysis. However, the diagnostic yield would be no less than 83% if all inflammation findings turned out to have a more specific diagnosis. A pilot study with the initial experience of Ion™ with the addition of the technology of limitation or attenuation of CT-to-body divergence described a correction of the divergences in 50% of the cases despite Ion™ considering a 100% success of navigation to the lesion [12]. After adjustments, the Cios Spin^®^ showed a tool within the lesion in 96.7% of cases. Two patients had hemodynamic instability and were excluded from the analysis since these procedures were interrupted. The diagnostic yield surpassed 93%. It is important to note that 73.3% of lesions were malignant, and such a high malignant rate might have contributed to this excellent diagnostic yield. O-ARM^®^ imaging (Medtronic, Minneapolis, MN, USA), which has 2D and 3D fluoroscopy technology like Cios Spin^®^, was also recently described to assist in RAB procedures [26]. A biopsy tool within the lesion was demonstrated in 77 cases (97%). The diagnostic yield for lesions ≤10 mm or 11–20 mm was 100% (7 lesions) and 78% (32 lesions), respectively. A recent retrospective study compared the use of EMN with digital tomosynthesis versus Ion™ even before the incorporation of its CT-to-body divergence correction system and showed no difference in diagnostic yield between the two technologies [27]. A retrospective, multicenter study compared the diagnostic yield of pulmonary nodules in 113 patients who underwent robotic-assisted biopsy (RAB) versus 112 patients who had CT-guided transthoracic biopsy (CTTB). The study found comparable diagnostic yields for malignant nodules in both groups, but with fewer complications in the RAB group. Additionally, the risk of pneumothorax was greater in the CTTB group [28].

### 4.2. Monarch™

The main studies to date utilizing this technology are summarized in Table 3. The first human study with this technology was a feasibility study, so it had a small number of patients and lesions [29]. Biopsy was possible in 14 of the 15 lesions. The initial larger-scale study using the Monarch system reported an average lesion size that was not particularly small [30]. Navigation success, defined as obtaining visualization with RP-EBUS or “diagnostic material”, occurred in 88.6% of cases, and material for cytopathological analysis was successfully collected in 98.8%. Another prospective feasibility study had navigation success based on visualization of the lesions with RP-EBUS in 51 of the 53 where the latter was used [31]. The diagnostic yield was 80.6% for concentric lesions versus 70% for eccentric ones in RP-EBUS. However, the overall diagnostic yield was not much better than described in the introduction. Despite being a single-center retrospective study, the work by Agrawal and collaborators is of excellent quality due to how diagnostic accuracy was used and the 12-month follow-up [32]. Despite the bronchus sign being present in 75% of cases, such a sign did not seem to have significantly influenced diagnostic accuracy which was around 85% when the lesion was visualized by RP-EBUS and 38% when this did not happen. It is worth mentioning that five patients did not have follow-up and these cases were considered without a diagnosis. The Monarch technology with the concurrent use of CBCT has also been described. CT was used in cases described as “extremely difficult” for bronchoscopy biopsy [33]. The authors reported that TIL or RP-EBUS was present within the lesion according to CT images in 100% of cases. The sensitivity for malignant disease was 86.6%. Of the seven cases without a diagnosis, five were reported to be confirmed benign on follow-up. Two cases required other invasive procedures.

In another retrospective study, 264 patients were analyzed across multiple centers, with the majority (over 99%) undergoing procedures using the Monarch platform in community hospitals [34]. The median size of the lesions observed was almost 20 mm. This study reported an “index” diagnostic yield of 85.2%. However, the authors stated the diagnostic yield could have varied from as low as 58.7% at the index procedure to as high as 89.0% based on different diagnostic yield definitions reported in the literature. The study also noted a slightly higher rate of pneumothorax along with chest tube placements when historically compared to other RAB studies.

### 4.3. Galaxy™

At the time of writing this manuscript, there were no published human studies employing this technology.

## 5. Final Considerations

Robotic-assisted bronchoscopy shows potential. Comparatively, the published diagnostic yields of two (Ion™ and Monarch™) of the available technologies on the market seem to surpass that of traditional peripheral bronchoscopy methods based on historical data. One study has shown that robotic bronchoscopy without image guidance (or target location updating) provides a similar yield to catheter-based navigational bronchoscopy that uses image guidance [27]. Currently, there are no human data yet available to evaluate the newer Galaxy™ system. However, the enhancement in diagnostic yield seems to be modest when traditional two-dimensional fluoroscopy is employed for guiding target sampling. We believe CT-to-body divergence plays a significant role in the limited improvement in diagnostic outcomes. The existing data shown here and reported by others support this view [14,15]. Primarily drawing from a limited number of studies that used the Ion™ platform along with its CT-to-body divergence reduction technology (via Cios Spin^®^ three-dimensional fluoroscopy or CBCT), the diagnostic yield seems to fall within the 80% range, and in some cases, even reaches the 90% range. And this was with median lesion sizes under 20 mm along the longest diameter. The Monarch™ platform is expected to have such technology available by 2024. The Galaxy™ system can incorporate commonly used fluoroscopy systems for target location updating, so there are expectations that it can reach similarly high diagnostic yield rates.

To the best of our knowledge, Cleveland Clinic may be the only center in the world that boasts expertise in all three currently available RAB platforms. In our experience, we have found that most of our proceduralists now choose to conduct their procedures utilizing the platforms that have available CT-to-body divergence correction capability. To have such technology with the Ion™, an additional investment to acquire the Cios Spin^®^ fluoroscopy device is necessary, while the Galaxy™ has this built-in functionally for use with commonly available two-dimensional fluoroscopy devices. This should be an important up-front cost advantage for the Galaxy™ if the Cios Spin^®^ is not already available at one’s institution. Now that we have CT-to-body divergence correction capabilities, we attempt sampling lesions that we never thought would be possible until 1–2 years ago, such as sub-centimeter (image) and near diaphragmatic (image) lesions, and we rarely use RP-EBUS when utilizing the Ion™ and Galaxy™ or with the Monarch™ when using CBCT. We also need to point out we do not know at present if either shape-sensing or EMN technologies will have any diagnostic advantage over the other. We do know delayed navigation when sampling the targets may impact the yields due to atelectasis development and further worsening CT-to-body divergence [10,16,17,18]. It is unclear whether positioning the fluoroscopy device before the initiation of navigation with shape-sensing technologies offers any advantage over EMN-based technologies in cases in which there is high risk of rapid atelectasis development.

While there is a growing interest in the potential benefits of RAB procedures, there remains a significant gap in research specifically addressing its cost-effectiveness. Early observations suggest that, due to lower complication rates associated with this procedure, there may be fewer hospitalizations needed to address these complications. A recent meta-analysis of RAB studies described that the pooled rate of pneumothorax was 0.60%, ranging between 0 and 9.4%, with <0.01% requiring chest tube placement [35]. In addition, simultaneous EBUS staging of the mediastinum can be performed along with RAB [6]. However, it is important to note that the increased costs associated with the specialized equipment required for RAB could impact on its cost-effectiveness. Despite the anticipated benefits, comprehensive studies are needed to calculate the cost-effectiveness of this procedure accurately. Such studies would provide a clearer understanding of the procedure’s financial implications in comparison to its clinical benefits.

There is also significant interest in RAB as a potential tool for the treatment of peripheral lesions due to its scope stability. A variety of ablation technologies (cryospray, laser, microwave, photodynamic therapy, radiofrequency) have been or are currently being investigated for endobronchial use and treatment of intrapulmonary lesions [36,37,38]. Therefore, RAB may facilitate the delivery of these and other treatment modalities directly into the peripheral tumor, especially when RAB is being utilized with target-updating technologies. Some believe that the combination of RAB, EBUS, and one or more of the ablation technologies cited here may be able to one day stage, diagnose, and treat lung cancer in one sitting.

In conclusion, we project that the diagnostic yield will, in due course, be comparable to CTTB. The diagnostic yield appears to be above 80% when RAB is used with CT-to-body divergence correction technologies. As mentioned at the beginning, there is no doubt that there is a selection bias of cases in all these studies. Therefore, without a comparative study of the various existing technologies, it will not be possible to determine a fundamental difference between them. Also, the heterogeneity in the definition of diagnostic yield brings even more difficulties in attempting comparisons. Finally, these technologies, when not integrated, add extra costs and may not be feasible for most centers worldwide. Regardless, robotic bronchoscopy is a welcome technology and should be seriously considered as an addition to any advanced diagnostic bronchoscopy center.

## Figures and Tables

**Figure 1 diagnostics-14-00399-f001:**
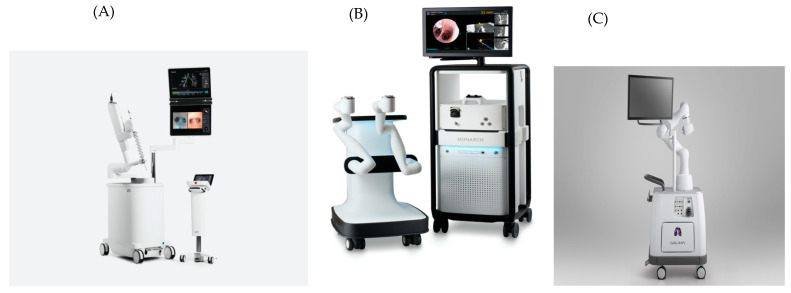
Ion™ platform (**A**; courtesy of Intuitive), Monarch™ platform (**B**; courtesy of Auris), and Galaxy platform from Noah Medical (**C**; courtesy of Noah Medical).

**Figure 2 diagnostics-14-00399-f002:**
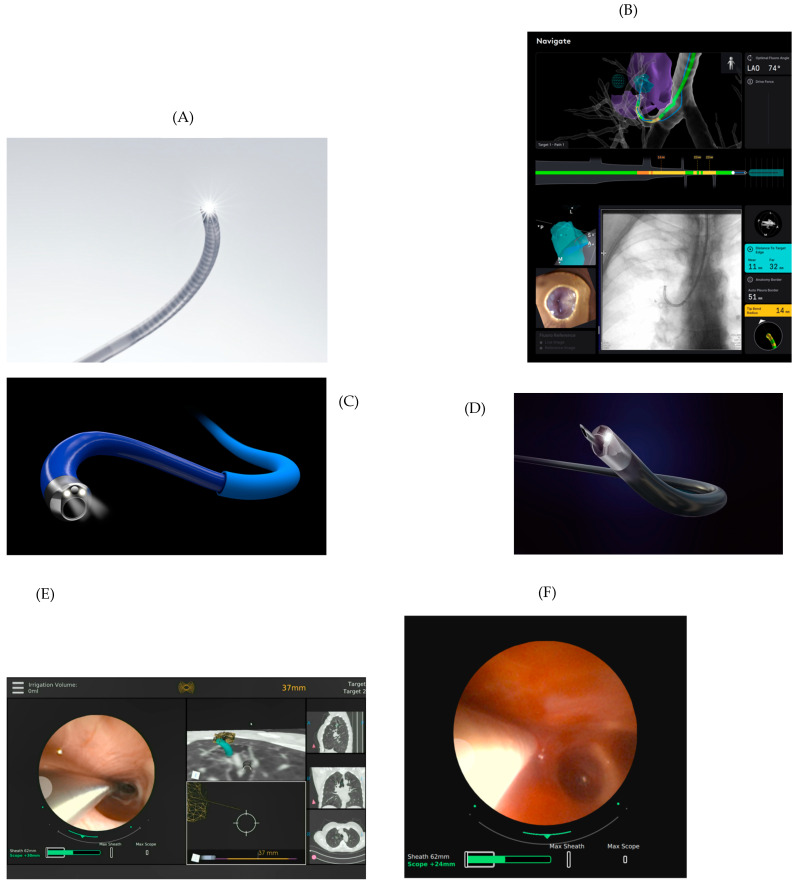
Ion™ catheter with camera (**A**; courtesy of Intuitive). Distal internal view of the same catheter when the camera is being removed (**B**, left lower corner). Also in panel B, one can observe some of the various images seen on the Ion™ screen when navigating to the lesion. Monarch™ bronchoscope with sheath (**C**; courtesy of Auris). Galaxy™ bronchoscopy with needle, camera, and light source (**D**; courtesy of Noah Medical). View of the RP-EBUS in the airway (image on the left) using the Monarch™ to adjust the ideal puncture point for biopsy based on ultrasonographic findings and guided by the location of the lesion with the help of the other five images (**E**). View of the airway using the Monarch™ at the puncture (**F**) moment.

**Figure 3 diagnostics-14-00399-f003:**
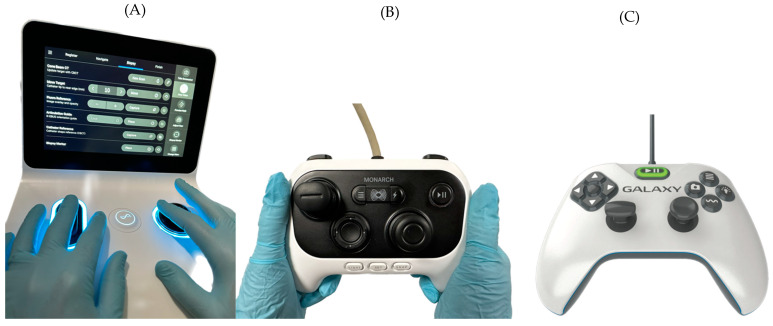
The Ion™ control has two buttons: the left to advance/retract the catheter and the right to move the distal end in any direction (**A**). Also, part of the control is a screen that functions like a smartphone to control the various functions of the platform. The Monarch™ and Galaxy™ controls look very similar to the Xbox video game controller (**B**,**C**). Like the Ion™ controller, the left button advances or retracts, and the right button moves the distal end of the catheter in any other direction. The other buttons are used at various moments during the procedure. At the front of the control are buttons for washing and suction and coupling/uncoupling (Monarch™) between the bronchoscope and the sheath.

**Figure 4 diagnostics-14-00399-f004:**
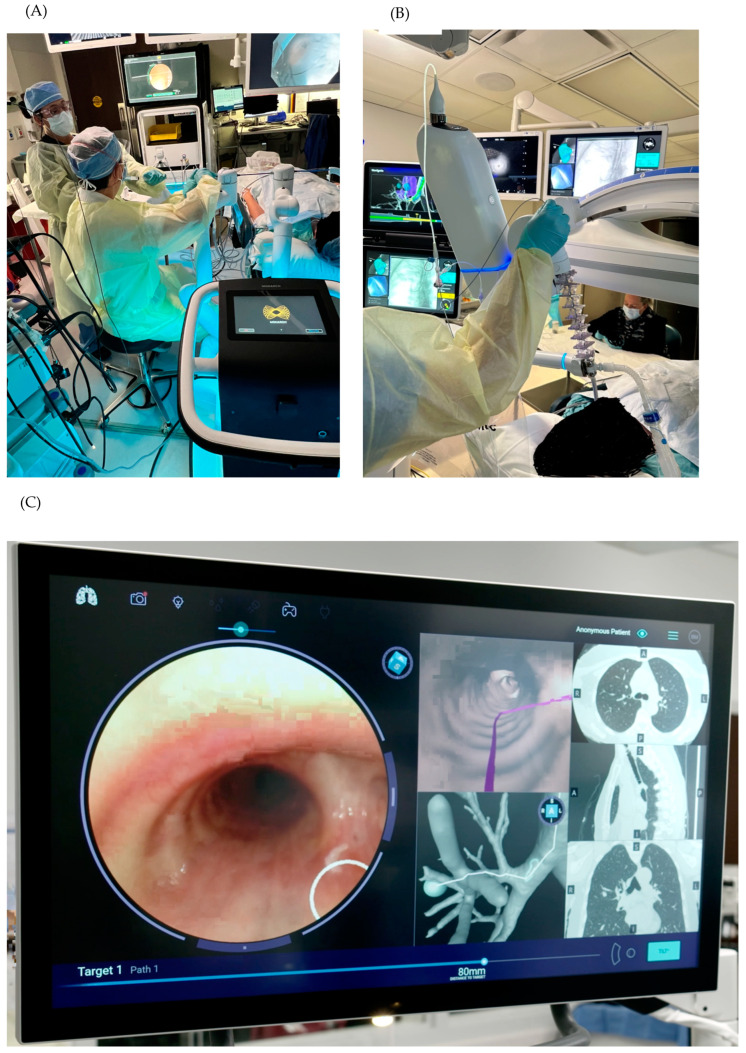
Image (**A**) shows the operator performing a biopsy with the Monarch™ platform. In panel (**B**), the operator is seen at the moment of biopsy through the Ion™ catheter. Panel (**C**) shows the Galaxy™ screen during navigation. The Galaxy™ platform during sampling utilizing its augmented fluoroscopy (**D**) and utilizing its TILT+^TM^ technology where the needle can be seen in the lesion (**E**).

**Table 1 diagnostics-14-00399-t001:** Notable features of robotic bronchoscopy technologies.

Robotic-Assisted Bronchoscopy Features	Ion™ Intuitive	Monarch™ Auris	Galaxy™ Noah Medical
Navigation technology	Shape sensing	Electromagnetic navigation	Electromagnetic navigation augmented fluoroscopy
Bronchoscopy	Catheter with removable vision probe (disposable after 5 uses) *	Sheath and bronchoscope with built-in camera (disposable after 2 uses) *	Bronchoscope with built-in camera (single use)
Catheter articulation	180°	180° (sheath: 130°)	180°
Catheter outer diameter	3.5 mm (catheter)	Outer sheath: 6 mm/inner scope: 4.2 mm	4.0 mm
Working channel diameter	2.0 mm	2.1 mm	2.1 mm
Irrigation and aspiration	No	Yes	Yes
CT-to-body divergence correction system	Yes (only with Cios Spin)	No	Yes
Augmented fluoroscopy	No	No	Yes
Tool-in-lesion confirmation	Yes (only when utilizing Cios Spin)	No	Yes
Tactile feedback	No	No	No

* Cleaning and recycling are carried out by Intuitive and Auris for their respective scopes.

**Table 2 diagnostics-14-00399-t002:** Key results from literature review up to December 2023 using the Ion™ platform.

Article: Author (Year)	Prospective versus Retrospective Single Center or Multicentric	Number of Lesions (Number of Patients)	Diagnostic Performance	Strict versus Lenient Diagnostic Performance Definition	X-ray Imaging	Mean Diameter of Lesions (mm)	Number of Lesions with Positive Bronchus Sign (%)	Number of Lesions in the Lower Lobes (%)	Number of Radial EBUS Confirmations (%)	Number of Pneumothorax (%) Number Requiring Drinage (%)
Fielding et al. (2019) [20]	Prospective Single center	29 (29)	79.3%	Not strict	Two-dimensional fluoroscopy	14.8	17 (58.6)	7 (24.1)	27 (93.1)	0
Simoff et al. (2021) [21]	Prospective Multicentric	67 (60)	Not described	N/A	Two-dimensional fluoroscopy	20	25 (37.3)	29 (43.2)	59 (89.4)	0
Benn et al. (2021) [22] *	Prospective Single center	59 (52)	86%	Less strict	CBCT	21.9	27 (46)	14 (23.7)	Not used	2 (3.8) 1 (1.9)
Kalchiem-Dekel et al. (2022) [23]	Prospective Single center	159 (131)	81.7%	Strict	Two-dimensional fluoroscopy in 79.9%: two- and three-dimensional fluoroscopy in 20.1%	18 (median)	100 (62.9)	54 (34)	124 (91.2)	2 (1.5) 2 (1.5)
Reisenauer et al. (2022) [24]	Prospective Multicentric	270 (241)	Not described	N/A	Not reported	18.8	ND	83 (30.9)	232 (86.6)	8 (3.3) 1 (0.4)
Styrvoky et al. (2022) [25]	Retrospective Single center	209 (198)	Not described 89% (calculated based on provided data; 83.2% if all inflammation findings had an alternative diagnosis)	Not strict	CBCT (98.6%)	22.6	126 (60.3)	67 (32.1)	183 (87.5)	2 (1) 1 (0.5)
Reisenauer et al. (2022) [12] †	Prospective Single center	30 (30)	93.3%	Strict	Three-dimensional fluoroscopy	17.5 (median)	12 (40)	9 (30)	23 (76.7)	0
Chambers et al. (2022) [26]	Retrospective Single center	79 (75)	77%	Strict	Three-dimensional fluoroscopy	20 (median)	44 (56)	23 (29)	Not used	2 (2.5) 1 (1.3)
Low et al. (2022) [27]	Retrospective Single center	143 (133)	77%	Strict	Two-dimensional fluoroscopy	17	57 (39.9)	51 (35.6)	127 (88.8)	2 (1.5) 2 (1.5)
Yu Lee-Mateus et al. (2023) [28]	Retrospective Multicentric	113 (113)	87.6% (no less than 81.4% if all inflammation cases with incorrect diagnosis)	Not strict	Two-dimensional fluoroscopy	ND	ND	35 (30.9)	ND	4 (3.5) ND

* Combined with cone-beam computed tomography. † Updated Ion™ technology, incorporating body-to-CT divergence elimination.

**Table 3 diagnostics-14-00399-t003:** Key results from literature review up to December 2023 using the Monarch™ platform.

Article: Author (Year)	Prospective versus Retrospective Single Center or Multicentric	Number of Lesions (Number of Patients)	Diagnostic Performance	Strict versus Lenient Diagnostic Performance Definition	X-ray Imaging	Mean Diameter of Lesions (mm)	Number of Lesions with Positive Bronchus Sign (%)	Number of Lesions in the Lower Lobes (%)	Number of Radial EBUS Confirmations (%)	Number of Pneumothorax (%) Number Requiring Drainage (%)
Rojas-Solano et al. (2018) [29]	Prospective Single center	15 (15)	Not described	N/A	Two-dimensional fluoroscopy	26	12 (80) *	7 (46.7)	Not used	0
Chaddha et al. (2019) [30]	Retrospective Multicentric	167 (165)	69.1 a 77% †	Strict	Two-dimensional fluoroscopy (presumed)	25	106 (63.5)	59 (35.3)	109 (89.4)	6 (3.6) 4 (2.4)
Chen et al. (2021) [31]	Prospective Multicentric	54 (54)	74.1% ‡	Not strict	Two-dimensional fluoroscopy (presumed)	23.2	32 (59.3)	17 (31.5)	51 (96.2)	2 (3.7) 1 (1.9)
Cumbo-Nacheli et al. (2022) [33]	Retrospective Single center	20 (20)	70% (calculated based on provided data)	N/A	CBCT	22	10 (50%)	6 (30)	15 (75)	0
Agrawal et al. (2023) [32]	Retrospective Single center	124 (124)	77% §	Strict	Two-dimensional fluoroscopy	24	93 (75%)	38 (30.6)	102 (82.2)	2 (1.6) 0 (0)
Khan et al. ^x^ (2023) [34]	Retrospective Multicentric	264 (264)	79.4% (at 12 months)	Not strict	Two-dimensional fluoroscopy (99.6%) CBCT (3.4%)	19.3 (median)	78/259 ^y^ (30.1%)	86 (32.6)	248 (93.9)	15 (5.7) 10 (3.8)

* Only peripheral lesions with bronchial signs were selected for this study. Therefore, the three lesions without bronchial signs were considered central. † Variation was estimated based on cases of untracked inflammation (N = 13), whether these findings were indeed correct or not, respectively. ‡ Calculated based on the data provided in the study. § Diagnostic accuracy, which has recently been suggested by some as possibly a superior way to evaluate the quality of these various techniques of peripheral bronchoscopy. ^x^ Monarch^®^ platform was used in 99.6% of patients. ^y^ Only 259 of 264 were evaluated/have available data of the presence of bronchus sign.

## Data Availability

Not applicable.

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
