# Peer review of "Robotic-Assisted Bronchoscopy: A Comprehensive Review of System Functions and Analysis of Outcome Data"

_diagnostics, 2024, doi:10.3390/diagnostics14040399_

Round 1

Reviewer 1 Report

Comments and Suggestions for Authors

The paper reviews currently available robotic-assisted bronchoscopy technologies. The topic is interesting, however, several issues suggest that the paper should be rearranged and completed.

First of all, the approach seems to be superficial, and a thorough survey of the state of the art is lacking. This consideration is also supported by a small number of references for a review article. Moreover, a general overview and interpretation are not provided by the authors. These issues are relevant in a review article to help the reader understand the proposed topic, the limitations and advantages of the current literature as well as future prospects.

Author Response

Dear Reviewer,

We would like to express our sincere gratitude for your insightful and constructive feedback regarding our manuscript on robotic-assisted bronchoscopy technologies. Your careful analysis and suggestions are extremely valuable for the enhancement of our work.

We would like to address a specific point mentioned in your comments: the perception that our article lacks a substantial number of references. Consequently, such a perception may lead to the thought that a more thorough survey of the state of the art is lacking. We recognize that at first, this may seem like an area of concern. However, we would like to highlight that the field of robotic-assisted bronchoscopy is relatively new and emerging. As a result, there is a limited amount of published literature available to date. This presented us with a unique challenge in conducting our comprehensive literature survey. Nevertheless, we acknowledge the importance of providing as complete a review as possible and are committed to improving our manuscript.

With this in mind, we propose the following actions for the revision of the article:

  • Expansion of Bibliographic Research/Explanation: We decided to increase some explanations about the location setting of the procedure, the sedation/anesthesia of the procedure, and a brief cost-effectiveness discussion.
  • Inclusion of Discussion on Future Perspectives: Given the emerging nature of the field, we will include a section dedicated to potential future directions, hypotheses, and unexplored areas that may inspire subsequent research.

We thank you again for your valuable contributions and hope that the proposed revisions meet your expectations and enrich our manuscript.

Reviewer 2 Report

Comments and Suggestions for Authors

The manuscript titled "Robotic-Assisted Bronchoscopy: A Comprehensive Review of System Functions and Analysis of Outcome Data" submitted to Diagnostics (diagnostics-2827035), describes update findings regarding robotic-assisted bronchoscopy technologies and discuss the existing scientific evidence supporting these. Review is very well organized and written. Manuscript is divided properly and after comprehensive introduction, authors described very well Robotic-assisted bronchoscopy technologies, including navigation, catheter/bronchoscope, controls, display screens Review is written very comprehensive way based on 33 actual and properly selected references. An author cconcluded at the end that  that robotic-assisted bronchoscopy shows potential and Cleveland Clinic will reach the clinic within the foreseeable future, probably the only center in the world that boasts expertise in all three currently available RAB platforms.

Author Response

Dear Reviewer,

We would like to express our sincere gratitude for your insightful and constructive feedback regarding our manuscript on robotic-assisted bronchoscopy technologies. Your careful analysis and suggestions are extremely valuable for the enhancement of our work.

Nevertheless, we acknowledge the importance of providing as complete a review as possible and are committed to improving our manuscript.

With this in mind, we propose the following actions for the revision of the article:

  • Expansion of Bibliographic Research/Explanation: We decided to increase some explanations about the location setting of the procedure, the sedation/anesthesia of the procedure, and the cost-effectiveness explanation.
  • Inclusion of Discussion on Future Perspectives: Given the emerging nature of the field, we have included a brief discussion on potential future directions in the final considerations section.

Reviewer 3 Report

Comments and Suggestions for Authors

Manuscript ID diagnostics-2827035

I congratulate the Authors for this exhaustive report about Robotic-Assisted Bronchoscopy.

The objective of the review was “to describe how RAB technology currently works and to discuss the current scientific evidence” and the Authors met the purpose with an exhaustive description of technology and comparison with Electromagnetic Navigation and Electromagnetic Navigation Augmented fluoroscopy.

The originality and the added value of the paper of the paper are the novelty of the technique that, as written by the Authors “shows potential” in terms of diagnostic yield. Future technology might improve CT-to-body divergence too, but this data needs to be confirmed.

Regarding the Conclusions, the costs of the procedure need to be discussed in detail analyzing the cost/benefit ratio of the procedure. Is this a procedure that can be performed as an outpatient one?

Is deep sedation required?

I think that the Authors should add the above information to the paper, adding, at the same time some literature about the procedure modality per se.

Author Response

Dear Reviewer,

We would like to express our sincere gratitude for your insightful and constructive feedback regarding our manuscript on robotic-assisted bronchoscopy technologies. Your careful analysis and suggestions are extremely valuable for the enhancement of our work.

Based on your feedback/review, we decided to include some topics to improve the quality of the review. With this in mind, we propose the following actions for the revision of the article:

  • Expansion of Bibliographic Research/Explanation: We decided to increase some explanations about the location setting of the procedure, the sedation/anesthesia of the procedure, and the cost-effectiveness explanation.
  • Inclusion of Discussion on Future Perspectives: Given the emerging nature of the field, we have included a brief discussion on potential future directions in the final considerations section.

We thank you again for your valuable contributions and hope that the proposed revisions meet your expectations and enrich our manuscript.

Round 2

Reviewer 1 Report

Comments and Suggestions for Authors

The Authors have properly addressed my concerns with the original manuscript.